# Inhibition of P-Glycoprotein Asymmetrically Alters the In Vivo Exposure Profile of SGC003F: A Novel Guanylate Cyclase Stimulator

**DOI:** 10.3390/ph17091140

**Published:** 2024-08-29

**Authors:** Jinle Lou, Nan Li, Xue Jiang, Xu Cai, Lingchao Wang, Xia Wu, Wenpeng Zhang, Chunmei Jin, Xiaomei Zhuang

**Affiliations:** 1Beijing Institute of Pharmacology and Toxicology, Beijing 100850, China; 15665741174@163.com (J.L.); 13345085440@163.com (N.L.); jiangx2021@163.com (X.J.); lj3232000@163.com (X.C.); 13426199839@163.com (L.W.); seancong1005@163.com (X.W.); wpzhang@bmi.ac.cn (W.Z.); 2College of Pharmay, Yanbian University, Yanji 133000, China; cmijin@ybu.edu.cn

**Keywords:** SGC003F, P-gp, drug–drug interactions, tissue distribution, excretion

## Abstract

As a novel guanylate cyclase stimulator, SGC003F is being developed for the treatment of heart failure with a reduced ejection fraction (HFrEF). This study aimed to assess the effect of P-glycoprotein (P-gp) inhibition on SGC003F exposure in vivo, comparing plasma and tissue levels, and evaluating the role of P-gp in the small intestine, blood–brain barrier (BBB), and kidney in impacting the tissue exposure. Tariquidar, a P-gp inhibitor, was added to monolayer transport assays to observe the changes in the transmembrane characteristics of SGC003F. Rats were given SGC003F with tariquidar via various routes to measure plasma, tissue, urine, and fecal concentrations. The inclusion of tariquidar significantly altered the pharmacokinetics of SGC003F. In LLC-PK1-MDR1 cells, tariquidar reduced the efflux ratio of SGC003F from 6.56 to 1.28. In rats, it enhanced the plasma AUC by 3.05 or 1.61 times, increased the Cmax by 2.13 or 1.07 times, and notably improved bioavailability from 46.4% to 95%. Additionally, co-administration with tariquidar led to a decrease in fecal excretion and an increase in tissue exposure, with only a moderate effect on the partition ratios in the small intestine and brain. P-gp inhibition impacts SGC003F exposure, with plasma levels not fully reflecting tissue levels. P-gp in the small intestine and BBB affects SGC003F’s pharmacokinetics, warranting further clinical drug–drug interaction (DDI) studies.

## 1. Introduction

Heart failure (HF) is a significant global public health issue characterized by high morbidity and mortality [1]. HF refers to the inability of the heart to pump enough blood to maintain the body’s metabolism needs, leading to symptoms such as dyspnea, limited activity, and fluid retention. In severe cases, symptoms such as shortness of breath, chest tightness, double-leg edema, dizziness, and may even be fatal [2]. Drug therapy remains effective for HFrEF [3]. However, patients receiving quadruple therapy still face the risk of worsening heart failure and death. Therapeutic drugs available for HFrEF do not fully meet clinical demands, and there is an urgent need for more effective drugs.

Recent studies have shown that the nitric oxide (NO)-soluble guanylate cyclase (sGC)-cyclic guanosine monophosphate (cGMP) pathway plays a crucial role in regulating cardiovascular function and serves as a novel target for the treatment of HF [4]. Riociguat, the inaugural sGC stimulator approved for clinical use [5], has seen restricted application in HF treatment due to its brief half-life, a consequence of metabolism by cytochrome P450 (CYP) enzymes [6]. Primarily, it is now prescribed for pulmonary arterial hypertension and chronic thromboembolic pulmonary hypertension [7]. The development of vericiguat, a derivative of riociguat with an extended half-life achieved through structural modification, marks a significant advancement [8]. Vericiguat is the first-in-class drug that targets the NO-sGC-cGMP signaling pathway for the treatment of HFrEF. Currently, clinical use of vericiguat is indicated for the treatment of adults with symptomatic chronic heart failure with an EF of less than 45% following hospitalization for heart failure or the need for outpatient intravenous diuretics [9]. Studies have shown that the combined use of vericiguat with standard therapy can significantly reduce the risk of cardiovascular death or hospitalization for heart failure in patients, providing a new approach to the treatment of heart failure [10,11].

SGC003F (chemical structure of which is depicted in Figure 1A) is a novel structural compound derived from the structural modification and screening of the lead compound vericiguat at our institute. Its preclinical efficacy assessment against HFrEF is superior to vericiguat. Considering that patients with HF typically require combination therapy in clinical settings and, due to the presence of co-morbidities such as hypertension, myocardial ischemia, atrial fibrillation, diabetes, and dyslipidemia, they need to receive multiple medications [12,13]. When multiple drugs are used in combination, besides the action of drug metabolic enzymes, P-gp is also an important cause of drug interactions [14]. Previous studies have found that vericiguat may be a substrate of P-gp [15]; however, no clinically significant differences were observed with co-administration of digoxin (P-gp substrate) [9,16].

Tariquidar, a potent, specific, noncompetitive P-glycoprotein inhibitor [17], has been noted in previous studies to potentially affect the pharmacokinetics of numerous drugs when co-administered. For example, the effect of tariquidar co-administration with loperamide on the pharmacokinetics and brain distribution of loperamide in rats: one hour after administration, 1.0 mg/kg of tariquidar increased loperamide levels in the brain by 2.3-fold [18].

The aim of this study is to elucidate the effects of drug-metabolizing enzymes and P-gp on SGC003F by employing an in vitro liver microsome incubation system and in vitro monolayer cell model expressing P-gp. Subsequently, the changes in plasma, major tissues, and excreta of SGC003F concentration in rats after co-administration with tariquidar will be investigated (the technical roadmap is shown in Figure 1B). The study will further focus on observing the alterations in plasma exposure levels during DDI to determine if these changes reflect variations in drug exposure across different tissues. Additionally, the contributions of P-gp in the intestine, BBB, and renal tubules, among other major organs, to the exposure to SGC003F will be compared.

## 2. Results

### 2.1. Effects of Tariquidar on Bidirectional Transport of SGC003F in Monolayer Cells

Digoxin was selected as the positive quality-control drug for the bidirectional transporter assay on Caco-2 cells, with the results shown in Appendix A. The average ER value of digoxin was 14.5, confirming the successful establishment of Caco-2 cells. The bidirectional transport rate of SGC003F across Caco-2 cells is also shown in Appendix A. The results indicated that there was a notable difference in the bidirectional permeation rate of SGC003F, with an ER value of 7.2, suggesting that SGC003F may act as a substrate for efflux transporters such as P-gp or breast cancer resistance protein (BCRP). To further investigate whether SGC003F is a substrate for P-gp, we utilized LLC-PK1-MDR1 cells to assess the transport characteristics of SGC003F.

The results of the digoxin as a P-gp substrate probe demonstrated (Figure 2A,B and Appendix A) that in LLC-PK1 MOCK cells (essentially not expressing MDR1), the ER value of digoxin remained around 1 in the presence or absence of tariquidar, with no significant alteration in permeability and efflux rate. However, in LLC-PK1 MDR1 cells, the ER value exceeded 2 (10.48), and the transit rate of digoxin from basolateral to apical side was significantly higher than that in the opposite direction. After the addition of tariquidar, the ER value of digoxin decreases from 10.48 to 0.91. Positive drug results indicated that the LLC-PK1 MOCK and MDR1 monolayers have been successfully established.

The results of the bidirectional transport of SGC003F are depicted in Figure 2C,D. The vectorial transport of SGC003F was observed on LLC-PK1-MDR1 cells compared with LLC-PK1-MOCK cells. The P_app B-A_ value was significantly higher than P_app A-B,_ with an efflux ratio (ER) value of 6.71 for SGC003F (Figure 2), indicating that P-gp mediated the efflux transport of SGC003F. In the presence of tariquidar, the P_app B-A_ value of SGC003F decreased significantly, while the P_app A-B_ value increased significantly, resulting in a decrease in the ER value to 1.28 (Appendix A). These findings suggest that SGC003F is a substrate for P-gp.

### 2.2. In Vitro Stability in Rat and Human Liver Microsomes

Positive results were consistent with empirical laboratory values, indicating that the incubation system is good and can be used for in vitro metabolic stability studies (Appendix A). Figure 3 illustrates the percentage of drug remaining over time at a final concentration of 1 µM for SGC003F when incubated with rat or human liver microsomes at various time intervals up to 60 min. The data demonstrate that there was no significant metabolic elimination of SGC003F throughout the incubation period, suggesting that the compound is relatively stable under these conditions. These findings imply that UGT and CYP enzymes may play a minimal role in the metabolic elimination of SGC003F, both in rats and humans.

### 2.3. Effects of Tariquidar on Plasma Pharmacokinetics of SGC003F in Rats

In a comprehensive examination of the impact of P-glycoprotein (P-gp) on the in vivo exposure of SGC003F in rats, a comparative pharmacokinetic was conducted following the co-administration of tariquidar through different routes. The mean plasma concentration–time profiles of SGC003F in rats subsequent to the combined intravenous or oral administration of tariquidar are depicted in Figure 4A,B. Upon co-administration of 7.5 mg/kg tariquidar via the *i.v.* route, no discernible influence on the plasma levels of SGC003F was observed across the various time points measured. Furthermore, no statistically significant differences were noted in AUC_0-t_, CL, or V of SGC003F when compared with the control group, as detailed in Table 1. Conversely, the oral co-administration of 15 mg/kg tariquidar yielded a pronounced impact on the plasma concentration of SGC003F at the 4 h mark post-administration, achieving statistical significance (*p* < 0.01). In comparison to the control group, the AUC_0-t_ for SGC003F was found to be elevated by 1.49-fold (*p* < 0.05), and CL was reduced by 32% (*p* < 0.05), as illustrated in Table 1.

The mean plasma concentration–time curves of orally administered SGC003F in rats after combined intravenous or oral administration of tariquidar are depicted in Figure 4C,D. Following the combination of SGC003F with intravenous administration of 7.5 mg/kg tariquidar, the plasma concentration of SGC003F was consistently higher than that of the control group at all time points, although statistical differences in the plasma concentration of SGC003F were only observed at individual time points (*p* < 0.05). The AUC_0-t_ was found to be 1.61 times that of SGC003F alone (*p* < 0.01), There was no significant difference in C_max_ compared with the control group, while F increased to 69.7%, which was 1.50 times that of the control group (Table 1). After the combination of SGC003F with oral administration of 15 mg/kg tariquidar, the plasma concentration of SGC003F was consistently higher than that of the control group at all time points (*p* < 0.01). Additionally, both the AUC_0-t_ and C_max_ of SGC003F were significantly elevated compared to those of the control group (Figure 4E,F), which were 3.05-fold and 2.13-fold that of the control group, respectively. Furthermore, the F increased to 95.1%, which is 2.05 times that of the control group (Table 1).

The above results suggest that tariquidar may primarily inhibit the function of intestinal P-gp, resulting in increased plasma exposure of SGC003F, and that the DDI between tariquidar and SGC003F may occur in the gastrointestinal tract.

### 2.4. Effects of Tariquidar on the Excretion of SGC003F via Feces and Urine in Rats

Concentrations of SGC003F in continuously collected urine and feces were measured following combined intravenous or oral administration of tariquidar to investigate the effects of intestinal and renal P-gp on the excretion of SGC003F via urine and feces. As depicted in Figure 5A,B, SGC003F was primarily excreted as a parent drug through the urinary and fecal pathways in rats after oral administration alone (Table 2). Within 120 h, the cumulative urinary excretion accounted for 11.9 ± 1.36%, the cumulative fecal excretion accounted for 59.5 ± 14.4%, with the total recovery rate of the parent drug representing 71.5 ± 15.1% of the administered dose. The results for urinary and fecal recovery of combined tariquidar are presented in Figure 5A–D. Compared with the control group, there was a statistically significant difference in the fecal excretion of SGC003F after co-administration of 15 mg/kg tariquidar orally or 7.5 mg/kg intravenously, mainly observed after 24 h (*p* < 0.05). However, the urinary excretion of SGC003F was statistically different mainly after 12 h (*p* < 0.05). As shown in Figure 5E,F, after combined oral administration of 15 mg/kg tariquidar, the cumulative fecal excretion of SGC003F in rats within 120 h decreased from 59.5% to 11.5% (*p* < 0.01). After combined intravenous administration of 7.5 mg/kg tariquidar, the cumulative fecal excretion of SGC003F reduced to 19.3% (*p* < 0.01). Compared with the control group, the cumulative urinary excretion of SGC003F in rats increased from 11.9% to 25.3% after combined oral administration of 15 mg/kg tariquidar for 120 h (*p* < 0.01), while increased to 18.6% after combined intravenous of 7.5 mg/kg tariquidar (*p* < 0.01). The cumulative fecal excretion of SGC003F in rats within 120 h was significantly decreased, and the cumulative urinary excretion of SGC003F in rats was significantly increased regardless of tariquidar combined with oral administration. The above results indicate that the kidney may not be the main target organ for the occurrence of DDI for tariquidar and SGC003F, while the small intestine appears to be the main target organ for the occurrence of DDI between the two drugs.

### 2.5. Effects of Tariquidar on the Tissue Distribution of SGC003F in Rats

As illustrated in Figure 6, we evaluated the effects of SGC003F combined with tariquidar administration on the distribution of SGC003F in major tissues including the heart, liver, kidney, brain, and small intestine. At the time point of 4 h (Figure 6A,B), the concentration of SGC003F in the brain was notably low after oral administration of SGC003F alone. However, co-administration of tariquidar, either orally or intravenously, significantly increased the concentration of SGC003F in the brain at this time point by 4.2-fold and 3.2-fold, respectively (Appendix A, *p* < 0.05). Also, at the time point of 4 h, the brain-to-plasma concentration ratio was twice that of the control group (Figure 6B, *p* < 0.05). At the time point of 24 h, the concentration of SGC003F in the brains of rats in all groups was below the limit of quantification. The AUC_0–24h_ and K_p2_ values of SGC003F in the brain were found to be consistent with the results at 4 h when compared with the control group.

After oral administration of SGC003F alone and in combination with oral or intravenous administration of tariquidar, there were no significant differences in small intestinal SGC003F concentration and the ratio of small intestinal-to-plasma concentration at each time point among the groups. Because of the large individual variation in SGC003F concentrations in the small intestine, the differences in AUC_0–24h_ and K_p2_ values of SGC003F in the small intestine from the control group were even more significant than the results at individual time points (Table 3, Appendix A).

As illustrated in the concentration–time curves (Figure 6A), after co-administration of tariquidar orally, there were significant differences in the concentrations of SGC003F in the kidneys and liver at the 1 h and 4 h time points compared to the control group (*p* < 0.05), but there were no differences in the kidney-to-plasma concentration ratio and liver-to-plasma concentration ratio at individual time points (Figure 6A,B). Additionally, at individual time points, there were no statistically significant differences in the concentration of SGC003F in the heart or the heart-to-plasma concentration ratio when tariquidar was administered via different routes compared to the control group (Figure 6A,B).

Comparing the tissue exposure and plasma exposure among different groups reveals that when co-administration of intravenous tariquidar is compared with the administration of SGC003F alone orally, the plasma exposure increased by 1.4 times, while the heart, liver, and kidneys showed an approximate increase of 1.3 times. In contrast, the intestine and brain exhibited a higher rate of increase than plasma exposure, with increases of 2.2 times and 2.4 times, respectively. The K_p_ values for the heart, liver, and kidneys were around 0.9 times that of the control group, whereas the K_p_ values for the intestine and brain increased by 1.6 times and 2.0 times, respectively.

Similarly, when comparing the exposure after co-administration of oral tariquidar with that after oral administration of SGC003F alone, the plasma exposure doubled, and the heart, liver, and kidneys also saw a two-fold increase. However, the intestine and brain showed an even higher rate of increase compared to plasma exposure, with increases of 3.2 times and 4.1 times, respectively. The K_p_ values for the heart, liver, and kidneys did not significantly change compared to the control group, while the K_p_ values for the intestine and brain increased by 2.1 times and 2.0 times, respectively. Moreover, the increase in exposure in all tissues and plasma after co-administration of oral tariquidar was higher than that observed after co-administration of intravenous tariquidar.

These findings confirm that the heart, liver, and kidney may not be the target organs for DDI concurrence between tariquidar and SGC003F. Instead, the small intestine and brain are the main target organs for DDI between tariquidar and SGC003F.

## 3. Discussion

Heart failure patients often have multiple comorbidities and require treatment with a variety of medications [19,20]. As an anti-heart failure drug, whether SGC003F will undergo DDIs in clinical use is a question that must be addressed.

The factors contributing to DDI primarily involve drug metabolizing enzymes and transporters [21,22,23]. Therefore, in vitro models are firstly employed to study whether drug metabolizing enzymes and transporters are involved in the in vivo processes of SGC003F. The results of metabolic stability experiments using rat and human liver microsomes indicated that CYP enzymes and UGT enzymes were not the primary factors influencing the metabolic clearance of SGC003F in vivo, making the likelihood of metabolism-based DDIs unlikely. Furthermore, bidirectional transmembrane transport experiments utilizing Caco-2 cells, LLC-PK-MOCK, and LLC-PK1-MDR1 cells demonstrated that SGC003F is a substrate of P-gp.

P-gp, as a crucial efflux transporter, plays a significant role in various processes such as drug absorption in the intestine, biliary excretion, crossing the blood–brain barrier (BBB) into brain tissue, and renal excretion [24,25,26], and is also one of the key factors contributing to DDI caused by combined medications [27,28,29,30]. Given that drug metabolizing enzymes are not responsible for the metabolic clearance of SGC003F, it can be inferred that the efflux function of intestinal P-gp is the main reason for the relatively low oral bioavailability of 46.4% for SGC003F in rats.

Currently, the assessment of DDIs typically involves two stages: in vitro studies and clinical studies [23,31]. However, there is still some bias in extrapolating the results of in vitro studies to in vivo situation, especially for transporter-mediated DDI [32]. Clinical trials not only have the issues of being time-consuming and labor-intensive but are also limited by ethics, allowing only the observation of changes in drug plasma concentrations [16,33]. Many previous studies have confirmed that transporter-based DDIs may primarily manifest at the tissue level; changes in plasma drug concentrations are not the important indicators, and changes in drug exposure in tissues may be the main cause of toxicity [34]. Therefore, in this study, the plasma PK, tissue distribution, and excretion of DDI of SGC003F in rats after co-administration of P-gp inhibitor (tariquidar) were studied. Our observations include multiple aims: (1) the effect of P-gp regulation on the overall exposure of SGC003F in vivo; (2) whether plasma DDI represents the DDI of tissue exposure in vivo; the contribution of P-gp in the intestine, BBB, and renal tubules to SGC003F in vivo exposure was compared.

In order to differentiate the between function of P-gp orientated in the gut and liver, we designed experiments to observe the impact on the in vivo exposure of SGC003F after oral or intravenous administration of tariquidar to inhibit P-gp function in different organs (Table 1). Compared with intravenous administration of SGC003F alone, the co-administration of oral tariquidar resulted in significant changes in AUC_0-t_ and CL. When compared with the group given SGC003F orally alone, the co-administration of oral tariquidar increased the AUC_0-t_ and C_max_ of SGC003F in rats by 3.05 times and 2.13 times, respectively, suggesting that after oral administration of tariquidar, the intestinal P-gp was mainly inhibited, leading to a significant increase in the absorption of SGC003F, with an oral bioavailability that could reach 95.1%. However, after co-administration of intravenous tariquidar, the AUC_0-t_ of SGC003F in rats increased by 1.61-fold (Figure 4E,F). Therefore, it can be concluded that the intestinal P-gp has a more pronounced effect on the absorption of SGC003F than its role in the liver. Although tariquidar is widely used as an inhibitor of P-gp both in in vitro and in vivo studies in animals, recent studies have found that it still suffers from poor specificity [35]. This lack of specificity is one of the challenges that hinder the study of transporters. The concentrations of tariquidar used in vitro and in vivo experiments in this study were designed according to the literature reports [36,37]. The pharmacokinetic data from this study reveal that the selected tariquidar dosage significantly enhanced the oral bioavailability of SGC003F, reaching an impressive 95.1%. This outcome confirmed the efficacy of the dose in blocking P-gp activity. However, we acknowledge that the absence of tariquidar concentration verification is a limitation of this study. Additionally, we cannot rule out the potential involvement of BCRP in the transport of SGC003F.

In further exploring the impact of P-gp inhibition on the tissue distribution and excretion of SGC003F, we also conducted an analysis of changes in drug distribution in major tissues, as well as urinary and fecal recoveries following co-administration of the two drugs. In the excretion experiments, both intravenous and oral co-administration of tariquidar increased the cumulative urinary excretion rate of SGC003F (Figure 5E), while significantly decreasing the cumulative fecal excretion rate of SGC003F (Figure 5F). However, the total cumulative urinary and fecal excretion of SGC003F after co-administration of the inhibitor was only 50% of that of the control group. This reduction may be attributed to a compensatory increase in metabolic clearance of SGC003F due to enhanced in vivo drug exposure following P-gp inhibitor co-administration. After co-administration of tariquidar via different routes, the exposure of SGC003F in the plasma, liver, kidney, brain, intestine, and heart was increased (Figure 6A). By calculating the total tissue distribution coefficient (K_p2_), whether the increase in tissue exposure was due to the increase in plasma exposure or due to the inhibition of tissue P-gp was observed, resulting in a disproportionate increase in tissue exposure relative to plasma levels. The results showed that, compared with monotherapy, the K_p2_ values of the liver, kidneys, and heart after combination therapy did not change significantly, while the K_p2_ values of the small intestine and brain increased markedly (Figure 6C). It can be inferred that the increase in exposure to SGC003F in the heart, liver, and kidney is correlated with the increase in plasma exposure. Conversely, the elevated exposure to SGC003F in the small intestine and brain may be attributed to the inhibition of P-gp function in intestinal epithelial cells and cerebral vascular endothelial cells [38]. The results also showed that although the twofold increase in the brain K_p2_ value after co-administration of the inhibitor, the K_P2_ value remains low, which may not increase the risk of SGC003F CNS toxicity. According to the above experimental results, the kidney may not be the key target of P-gp-induced DDI, and the intestine and BBB may be the main tissues in which DDIs occur. These results underscore the importance of our findings for preclinical safety and toxicological assessments, particularly in highlighting the need to consider the potential toxicity of the drug to the central nervous system and gastrointestinal tract.

## 4. Materials and Methods

### 4.1. Reagents, Instruments and Cell Culture

Reagent material: SGC003F [39] (purity: 98.0%) and riociguat (purity: 98%, internal standard) were synthesized in house. Dimethyl sulfoxide (DMSO, purity > 99.7%) was purchased from Sigma-Aldrich (St. Louis, MO, USA). HPLC-grade acetonitrile and methanol were purchased from Thermo Fisher Scientific (Waltham, MA, USA). HPLC-grade formic acid was purchased from J&K Scientific (Beijing, China). Pure water was purchased from Hangzhou Wahaha Group Co., Ltd. (Hangzhou, China). Surfactant isopropyl myristate, polyoxyethylene 35 castor oil, and diethylene glycol monoethyl ether were obtained from Shanghai Xinyu Biotechnology Co., Ltd. (Shanghai, China). Tariquidar (Tar) was obtained from Shanghai Yuanye Biotechnology Co., Ltd. (Shanghai, China). Digoxin derived from Selleck Chemicals (Houston, TX, USA). Fetal bovine serum (FBS), Penicillin-streptomycin (PS, 10,000 U/mL), Dulbecco’s modified essential medium glutamax (DMEM), and Roswell Park Memorial Institute (RPMI) 1640 medium were purchased from Gibco (Waltham, MA, USA). Hanks’ balanced salt solution was purchased from Solarbio (Beijing, China). Rat and human liver microsomes were purchased from BioIVT (Carlsbad, CA, USA). Transwell cell culture plates and inserts were purchased from Corning Life Sciences (Tewksbury, MA, USA).

Instruments: The HPLC (30AD) and triple quadrupole tandem mass spectrometer (8060) were purchased from Shimadzu (Shimadzu, Kyoto, Japan). The high-speed refrigerated centrifuge (FRESCO21) was purchased from Thermo (Waltham, MA, USA). The electronic analytical balance (T214) was from Denver Instrument (Denver, CO, USA). The metabolic cages were purchased from UGO (Gemonio, Italy). The cell resistance meter was purchased from Beijing Metalwork Hongtai Technology Co., Ltd. (Beijing, China).

Cell lines: LLC-PK1 cells with stable-expressing human MDR1 and mock-transfected with empty vector (LLC-PK1-MOCK) were gifted by the lab of Professor Qingcheng Mao (University of Washington, Seattle, WA, USA). The culture medium of the LLC-PK1-MOCK and LLC-PK1-MDR1 cells was 1640 medium containing 10% FBS and 100 U/mL PS. The Caco-2 cells were purchased from ADCC. The Caco-2 cell culture medium was DMEM high-glucose medium (containing 10% FBS, 1% non-essential amino acid solution, 100 U/mL PS). Both cell lines were cultured at 37 °C in a cell incubator containing 5% CO_2_.

### 4.2. Experimental Animals and Drug Solution Preparation

Adult Sprague Dawley rats (230–280 g) were obtained from Beijing Vital River Laboratory Animal Technology Co., Ltd. (Beijing, China). The animals were kept in a controlled environment with a 12 h light/dark cycle and stable temperature, and had free access to food and water, adapting for one week. All animal procedures were approved by the Experimental Animal Ethics Committee of Beijing Institute of Pharmacology and Toxicology (IACUC-DWZX-2023-P718).

SGC003F and tariquidar were dissolved in a solvent consisting of 11% isopropyl myristate, 74% polyethylene glycol 35 castor oil, and 15% diethylene glycol monoethyl ether. The stock solution concentrations were 1 mg/mL for SGC003F and 22.5 mg/mL for tariquidar. Next, the stock solution of SGC003F and tariquidar were diluted with saline at a ratio of 1:9 and 1:3.5, respectively, to achieve a working solution of SGC003F (0.1 mg/mL) and tariquidar (5 mg/mL) for oral and intravenous administration in rats.

### 4.3. In Vitro Experiments

#### 4.3.1. Bidirectional Transport Experiments of Caco-2 and LLC-PK1 MOCK/MDR1 Monolayer Cells

LLC-PK1-MDR1, LLC-PK1-MOCK, and Caco-2 cells were seeded in 24-well Transwell cell plates at optimized densities. The medium was changed every other day for the LLC-PK1 cells and every day for the Caco-2 cells. The LLC-PK1 cells were cultured continuously for 4–7 days, while the Caco-2 cells were cultured continuously for 28–35 days in preparation for transport experiments. Prior to the experiment, TEER (transepithelial electrical resistance) values were measured using a cell resistance meter. For electrical measurements, two electrodes were used, with one electrode placed in the upper compartment and the other in the lower compartment and the electrodes were separated by the cellular monolayer. The measurement procedure included measuring the blank resistance (R_blank_) of the semipermeable membrane only (without cells) and measuring the resistance across the cell layer on the semipermeable membrane (R_total_). TEER values were achieved by calculating as TEER = (R_total_ − R_blank_) × M_area_ in the unit of Ω·cm^2^. The monolayer integrity was guaranteed by determining the TEER before and after the bidirectional transport experiments. The transport experiment could be conducted once the LLC-PK1 cells demonstrated a TEER value exceeding 70 Ω·cm^2^, and the Caco-2 cells surpassed 1200 Ω·cm^2^. After washing the cells with HBSS, the transport experiments were carried out in the apical-to-basal (A-B) or basal-to-apical (B-A) directions. For the A-B experiments, a solution of HBSS/Hepes/0.1% BSA buffer containing 2 µM SGC003F was added to the apical side of the Transwell, while blank HBSS/Hepes/1% BSA buffer was added to the basal side as the receiving solution. Conversely, in the B-A experiments, blank HBSS/Hepes/1% BSA buffer was added to the top side of the Transwell as the receiving solution, and HBSS/Hepes/0.1% BSA buffer containing 2 µM SGC003F was added to the bottom side as the feeding solution. In addition to SGC003F, a positive control group using digoxin was also included. The buffer solution for the P-gp inhibition group contained 5 µM tariquidar, which was co-incubated with SGC003F or Digoxin.

After administration, the Transwell chambers were placed into the incubator and incubated at 37 °C for 2 h. At time points of 0.5 and 1 h, 50 µL, samples were removed from the receiving side, followed by immediate replenishment with an equal volume of HBSS solution (supplemented with 10 mM Hepes and 1% BSA) to maintain the system’s integrity and continue the incubation process. Upon completion of the 2 h incubation, 100 µL samples were taken from both the donor and receiving side. All samples were promptly stored at −40 °C for analysis. Each experiment was repeated three times within each group.

#### 4.3.2. In Vitro Incubation in Rat and Human Liver Microsomes

In vitro incubation dependent on NADPH: SGC003F was added into rat and human liver microsomes diluted in PBS (100 mM, pH 7.3). NADPH was introduced into the system. The NADPH had been pre-equilibrated under the same conditions for 5 min to initiate the enzymatic reaction. The incubation mixture was meticulously prepared to achieve the following final concentrations: 1 µM for SGC003F, 0.5 mg/mL for the liver microsomal protein, and 1 mM for NADPH.

In vitro incubation dependent on UDPGA: rat and human liver microsomes diluted in PBS (100 mM, pH 7.3) were mixed with alamethicin and then placed on ice for 15 min before the addition of the working solution of SGC003F. After preincubation in a water bath at 37 °C for 5 min, UDPGA, which had been preincubated for 5 min was added to initiate the reaction. The final concentrations in the incubation system were as follows: 1 μM for SGC003F, 0.5 mg/mL for liver microsomal protein concentration, 5 mM for UDPGA, and 25 μg/mL for alamethicin.

Positive probe substrate assays were utilized to ensure normal activity of the incubation system. The total incubation period was 60 min, with sampling conducted at 0, 5, 15, 30, and 60 min (a 20 μL incubation sample was mixed into 180 μL ice-cold acetonitrile containing 5 ng/mL internal standard riociguat) to observe the drug’s changes over time and to calculate its in vitro half-life. Positive probe substrate assays were utilized to ensure normal activity of the incubation system. The reaction-terminated incubation samples were vortexed for 1 min and then centrifuged at 4 °C and 14,000× *g* for 10 min. Finally, 60 μL supernatant was taken and diluted with 60 μL of 50% acetonitrile before being analyzed by LC-MS/MS.

### 4.4. Pharmacokinetic Study in Rats

#### 4.4.1. Effect of Tariquidar on Plasma Pharmacokinetics of SGC003F in Rats

Thirty-six male SD rats were randomly divided into 6 groups, with 6 rats in each group. Group I: SGC003F (*p.o.*, 1 mg/kg) + blank vehicle (control group); Group II: SGC003F (*p.o.*, 1 mg/kg) + tariquidar (*i.v.*, 7.5 mg/kg); Group III: SGC003F (*p.o.*, 1 mg/kg) + tariquidar (*p.o.*, 15 mg/kg); Group IV: SGC003F (*i.v.*, 1 mg/kg) + blank vehicle; Group V: SGC003F (*i.v.*, 1 mg/kg) + tariquidar (*i.v.*, 7.5 mg/kg); Group VI: SGC003F (*i.v.*, 1 mg/kg) + tariquidar (*p.o.*, 15 mg/kg). Oral administration required fasting for 12 h in advance and free access to water. At 0.033 (*i.v.*), 0.083, 0.25, 0.5, 1, 2, 4, 8, 12, and 24 h after administration, 100 μL of blood was collected intravenously and placed in heparin tubes. All blood samples were centrifuged at 2500× *g* for 10 min at 4 °C, and plasma was stored at −40 °C for analysis.

#### 4.4.2. Effect of Tariquidar on the Excretion of SGC003F via Feces and Urine in Rats

In total, 18 male SD rats were randomly divided into three groups, with 6 rats in each group. Group I: SGC003F (*p.o.*, 1 mg/kg) + blank vehicle (control group); Group Ⅱ: SGC003F (*p.o.*, 1 mg/kg) + tariquidar (*i.v.*, 7.5 mg/kg); Group III: SGC003F (*p.o.*, 1 mg/kg) + tariquidar (*p.o.*, 15 mg/kg). The rats were fasted for 12 h before oral administration. After administration, the rats were placed immediately in a metabolic cage and resumed feeding for 2 h. Urine and fecal samples were collected at 0–4, 4–8, 8–12, 12–24, 24–36, 36–48, 48–72, 72–96, and 96–120 h after administration. Fecal samples were homogenized with acetonitrile at a ratio of 1:4 (*w*/*v*) and stored at −40 °C for analysis.

#### 4.4.3. Effect of Tariquidar on the Tissue Distribution of SGC003F in Rats

A total of 45 male SD rats were randomly divided into 15 groups, with 3 rats in each group; each set of 5 groups constituted a larger group. The first large group served as the control group: SGC003F (*p.o.*, 1 mg/kg) + blank solvent; the second large group was designated as experimental Group I: SGC003F (*p.o.*, 1 mg/kg) + tariquidar (*i.v.*, 7.5 mg/kg); the third group was labeled as Group II: SGC003F (*p.o.*, 1 mg/kg) + tariquidar (*p.o.*, 15 mg/kg). The rats in each group were euthanized at 0.5 h, 1 h, 2 h, 4 h, and 24 h after administration. Blood samples were collected by cardiac puncture and placed in heparin tubes. The plasma was harvested by centrifugation at 2500× *g* for 10 min at 4 °C. Major organs (brain, heart, liver, kidney, and small intestine) were collected and homogenized with saline at a ratio of 1:4 (*w*/*v*). All samples were stored at −40 °C for further analysis.

### 4.5. Sample Processing and LC-MS/MS Analysis

Biological samples were processed by the protein precipitation method. An amount of 20 μL of biological matrix sample was mixed with 20 μL acetonitrile, followed by the addition of 160 μL acetonitrile containing the internal standard riociguat (5 ng/mL). The mixture was then vortexed and mixed for 1 min, before being centrifuged at 14,000× *g* for 10 min. An amount of 60 μL of supernatant was quantitatively transferred into a tube containing 60 μL of pure water with 50% acetonitrile for further analysis.

The concentration of SGC003F in the biological matrix was quantified by LC-MS/MS. Shim-pack GIST C18-AQ columns (2.1 mm × 50 mm, 5 μm) were used by being kept at 40 °C. The mobile phase consisted of an A phase containing 0.1% formic acid water and a B phase containing 0.1% formic acid acetonitrile. The flow rate was 0.8 mL/min. The gradient elution was as follows: 0.0 to 0.3 min, 10% B; 0.3 to 2.5 min, 10–95% B; 2.5 to 3 min, 95% B; 3 to 3.1 min, 10% B; 3.1 to 4 min, 10% B. Mass spectrometry conditions included electrospray ionization (ESI) ion source, multiple reaction monitoring (MRM) mode, and positive ion scanning for quantitative analysis. The characteristic ion pairs were *m*/*z* 433.1→115 for SGC003F and *m*/*z* 423.1→109.1 for IS, respectively. The injection volume for tissue homogenate samples, liver microsomal incubation solution, and cell samples was 5 μL. The injection volume for plasma samples, feces, and urine samples was 2 μL.

### 4.6. Data Processing and Statistical Analysis

The main pharmacokinetic parameters, such as time to peak (t_max_), half-life (t_1/2_), peak concentration (C_max_), and area under the curve (AUC), bioavailability (F), were calculated using a non-compartmental model with WinNonlin 8.1 software (Princeton, NJ, USA).

The elimination rate constant (k_e_ or λ_z_) was determined from the terminal phase of the concentration–time curve by using a semilogarithmic plot. Furthermore, the half-life (t_1/2_) was calculated according to this equation
t_1/2_ = 0.693/k_e_(1)

Bioavailability (F) was calculated according to Equation (2):F = Dose(*i.v.*) × AUC(*p.o.*)/Dose(*p.o.*) × AUC(*i.v.*) × 100%(2)

The tissue plasma partition coefficient at individual time point (K_P1_) was calculated by the ratio of the tissue concentration to the plasma concentration at each time point, as described in Equation (3)
K_P1_ = C_tissue_/C_plasma_(3)

The overall tissue plasma partition coefficient (K_P2_) was calculated by the ratio of the area under the tissue drug-time curve to the area under the plasma drug–time curve, as described in Equation (4)
K_P2_ = AUC_(0-t, tissue)_/AUC_(0-t, plasma)_(4)

The apparent permeability rate value (P_app_) of the monolayer cells was calculated according to Equation (5)
P_app_ = (dQ/dt)/(A × C_0_)(5)
where dQ/dt is the transport rate (µmol·s^−1^); A is the insert surface area (0.33 cm^2^); C_0_ is the initial concentration of the compound (µM).

Efflux rate (ER) was calculated according to Equation (6):ER = P_app B-A_/P_appA-B_(6)

Statistical analyses were performed using IBM SPSS Statistics 26. The experimental data were expressed as mean ± standard deviation (SD). Shapiro–Wilk test was firstly utilized to test the normal distribution of the data. When exploring differences between multiple groups, a one-way ANOVA was used to determine if there was a significant difference between group means. For the comparison of the two groups of data, the independent sample *t*-test was used for statistical analysis. *p* value less than 0.05 was set as the threshold of significance. When *p* < 0.05, the observed difference was judged to be statistically significant.

## 5. Conclusions

In summary, based on the finding that P-gp is the primary factor influencing the pharmacokinetic behavior of SGC003F in vivo, experiments in rats has demonstrated that co-administration of the P-gp inhibitor can alter the plasma exposure of SGC003F. Additionally, the tissue distribution and excretion experiments revealed that changes in plasma drug concentration did not completely correspond with changes in tissue drug concentrations, with variations in the intestine and brain exceeding those in plasma. This suggests that P-gp in the intestine and BBB are key factors influencing the pharmacokinetics of SGC003F. In vivo DDI studies at the animal level offer significant advantages over human experiments, which will provide valuable insights for the in-depth investigation of DDIs.

## Figures and Tables

**Figure 1 pharmaceuticals-17-01140-f001:**
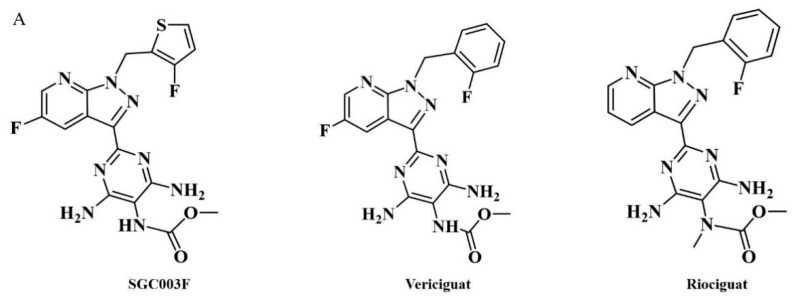
Chemical structures of SGC003F, vericiguat, and riociguat (**A**), and technical roadmap (**B**).

**Figure 2 pharmaceuticals-17-01140-f002:**
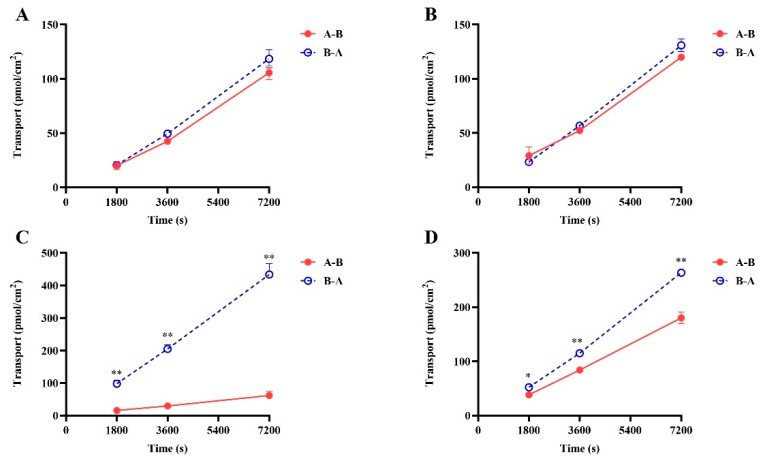
The bi-directional transport of SGC003F in LLC-PK1-MOCK (**A**,**B**) and LLC-PK1-MDR1 (**C**,**D**) monolayer cells. (**A**,**C**) SGC003F (2 μM). (**B**,**D**) SGC003F (2 μM) + Tar (5 μM). Data are expressed as mean ± SD (* *p* < 0.05; ** *p* < 0.01 compared with control; *n* = 3). Notes: A—apical, B—basolateral. A-B indicates absorption of the drug and B-A indicates efflux of the drug.

**Figure 3 pharmaceuticals-17-01140-f003:**
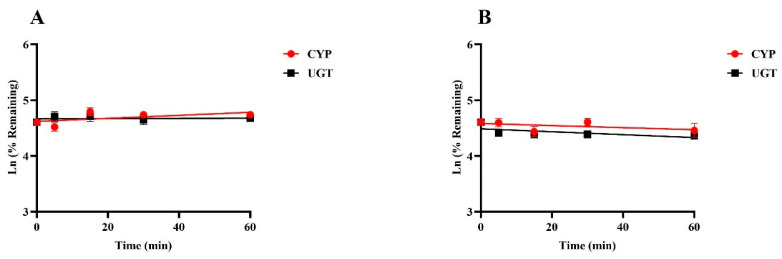
Metabolic stability of SGC003F in rat liver microsomes (**A**) and in human liver microsomes (**B**). Data are expressed as mean ± SD. (*n* = 3).

**Figure 4 pharmaceuticals-17-01140-f004:**
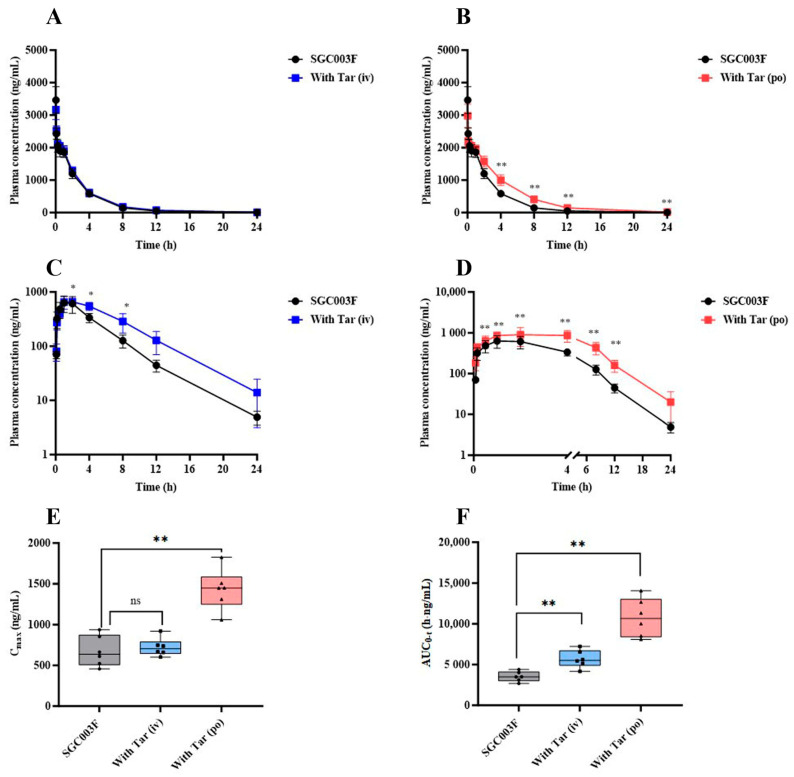
The alternations of SGC003F plasma pharmacokinetic behavior after combination with the P-gp inhibitor (tariquidar) via different dosing routes. The plasma concentration–time curve of SGC003F administered intravenously in rats with or without intravenous administration of 7.5 mg/kg tariquidar (**A**) or oral administration of 15 mg/kg tariquidar (**B**). The plasma concentration–time curve of SGC003F administered orally in rats with or without intravenous administration of 7.5 mg/kg tariquidar (**C**) or oral administration of 15 mg/kg tariquidar (**D**). (**E**) The AUC of SGC003F with or without tariquidar. (**F**) The C_max_ of SGC003F with or without tariquidar. Data are expressed as mean ± SD (ns *p* > 0.05; * *p* < 0.05; ** *p* < 0.01 compared with control; *n* = 6). Abbreviations: Tar = tariquidar.

**Figure 5 pharmaceuticals-17-01140-f005:**
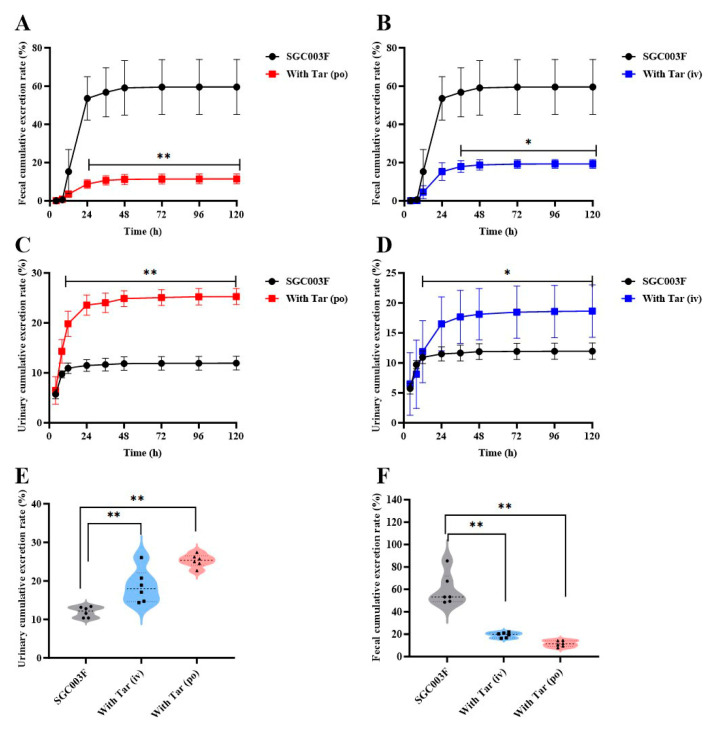
Urinary and fecal cumulative excretion of SGC003F in rats after combination of P-gp inhibitor (tariquidar) via different dosing routes. Fecal cumulative excretion rate of SGC003F in rats with or without oral administration of 15 mg/kg (**A**) or intravenous administration of 7.5 mg/kg Tar (**B**). Urinary cumulative excretion rate of SGC003F in rats with or without oral administration of 15 mg/kg (**C**) or intravenous administration of 7.5 mg/kg Tar (**D**). (**E**) Urinary cumulative excretion rate of SGC003F and SGC003F combined with tariquidar after 120 h. (**F**) Fecal cumulative excretion rate of SGC003F and SGC003F combined with tariquidar after 120 h. Data are expressed as mean ± SD (* *p* < 0.05; ** *p* < 0.01 compared with control; *n* = 6).

**Figure 6 pharmaceuticals-17-01140-f006:**
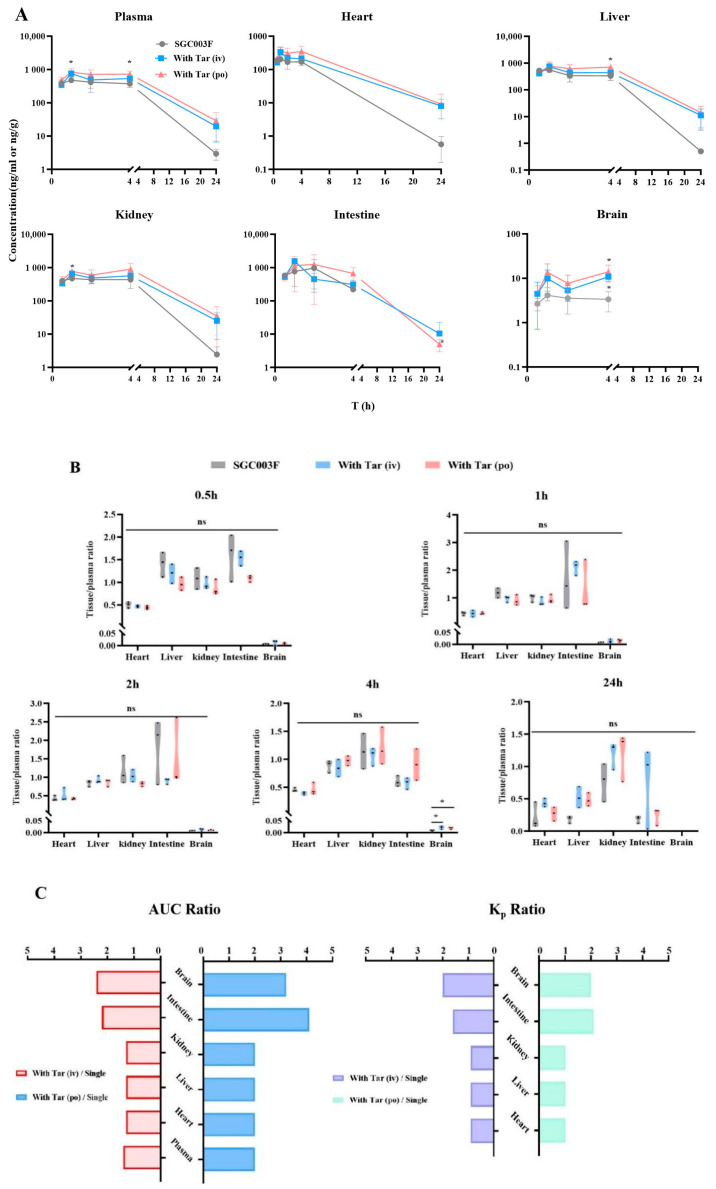
Mean biodistribution profiles and plasma concentration–time curves (**A**) and partitions of tissue to plasma at 0.5 h, 1 h, 2 h, 4 h, and 24 h after oral administration of 1 mg/kg SGC003F alone, combined with intravenous administration of 7.5 mg/kg or oral administration of 15 mg/kg tariquidar (**B**). Ratios of tissue and plasma AUC after dosing combined intravenous or oral tariquidar versus oral administration of SGC003F alone; K_p2_ ratios after dosing combined intravenous or oral administration of tariquidar versus oral administration of SGC003F alone (**C**). Data are expressed as mean ± SD (ns *p* > 0.05; *****
*p* < 0.05 compared to control; *n* = 3).

**Table 1 pharmaceuticals-17-01140-t001:** Pharmacokinetic parameters of SGC003F with or without tariquidar via different routes (*n* = 6, Mean ± SD).

Administration Route	Parameter	Unit	Single	With Tar (*i.v.*)	Ratio 1	With Tar (*p.o.*)	Ratio 2
*i.v.*	CL	mL/h/kg	131 ± 10.3	122 ± 8.66	0.93	89.2 ± 13.6 **	0.68
	V_ss_	mL/kg	419 ± 31.8	422 ± 32.8	1.00	382 ± 26.6	0.91
	AUC_0-t_	h·ng/mL	7625 ± 560	8168 ± 530	1.07	11,338 ± 1543 **	1.49
*p.o.*	t_1/2_	h	3.45 ± 0.46	4.31 ± 0.91	1.25	4.35 ± 0.35 **	1.26
	T_max_	h	1.33 ± 0.52	1.50 ± 0.55	1.13	2.33 ± 1.37	1.75
	C_max_	ng/mL	673 ± 187	722 ± 111	1.07	1433 ± 250 **	2.13
	AUC_0-t_	h·ng/mL	3539 ± 610	5694 ± 1079 **	1.61	10,781 ± 2347 **	3.05
	F	%	46.4	69.7	1.50	95.1	2.05

Ratio 1: with Tar (*i*.*v*.)/single; Ratio 2: with Tar (*p*.*o.*)/single; ** *p* < 0.01 (compared to control group).

**Table 2 pharmaceuticals-17-01140-t002:** Urinary and fecal cumulative excretion of SGC003F in rats with or without tariquidar via different routes (*n* = 6, Mean ± SD).

Parameters	Unit	Single	With Tar (*i.v.*)	Ratio 1	With Tar (*p.o.*)	Ratio 2
U	%	11.9 ± 1.36	18.6 ± 4.37	1.56	25.3 ± 1.62	2.13
F	%	59.5 ± 14.4	19.3 ± 2.31 **	0.32	11.5 ± 2.61 **	0.19
U + F	%	71.5 ± 15.1	37.9 ± 5.99 **	0.53	36.7 ± 3.36 **	0.51

Ratio 1: with Tar (*i*.*v*.)/single; Ratio 2: with Tar (*p*.*o*.)/single; ** *p* < 0.01 (compared to control group).

**Table 3 pharmaceuticals-17-01140-t003:** Tissue distribution and tissue plasma ratio of SGC003F in rats with or without oral administration (15 mg/kg) or intravenous administration (7.5 mg/kg) of tariquidar (*n* = 3, Mean ± SD).

Parameters	Tissue	Single	With Tar *(i.v.*)	Ratio 1	With Tar (*p.o.*)	Ratio 2
	Plasma	5308 ± 798	7545 ± 1277	1.4	10,221 ± 2269	2.0
	Heart	2370 ± 346	3056 ± 619	1.3	4747 ± 1822	2.0
AUC_(0–24h)_	Liver	4871 ± 1049	6365 ± 283	1.3	9599 ± 1821	2.0
(h·ng/mL)	Kidney	5993 ± 2167	7868 ± 1566	1.3	11,930 ± 4520	2.0
	Intestine	2533 ± 988	5629 ± 1361	2.2	10,398 ± 3752	3.2
	Brain	13.1 ± 2.66	31.2 ± 10.9	2.4	41.5 ± 14.8	4.1
K_P2_	Heart	0.45 ± 0.04	0.40 ± 0.02	0.9	0.45 ± 0.09	1.0
	Liver	0.91 ± 0.09	0.86 ± 0.11	0.9	0.95 ± 0.07	1.0
	Kidney	1.11 ± 0.24	1.05 ± 0.14	0.9	1.16 ± 0.26	1.0
	Intestine	0.47 ± 0.11	0.74 ± 0.06 *	1.6	1.01 ± 0.19 *	2.1
	Brain	0.002 ± 0	0.004 ± 0 *	2.0	0.004 ± 0 *	2.0

Ratio 1: with Tar (*i*.*v*.)/single; Ratio 2: with Tar (*p*.*o*.)/single; * *p* < 0.05 (compared to control group).

## Data Availability

The authors confirm that the data supporting the findings of this study are available within the article.

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
