# Peer review of "Inhibition of P-Glycoprotein Asymmetrically Alters the In Vivo Exposure Profile of SGC003F: A Novel Guanylate Cyclase Stimulator"

_pharmaceuticals, 2024, doi:10.3390/ph17091140_

Round 1

Reviewer 1 Report

Comments and Suggestions for Authors

First of all, congratulations to the authors who have provided a comprehensive study on the potential drug-drug interactions (DDIs) involving SGC003F. However, to further enhance the clarity and focus of your manuscript, I have several questions and suggestions that need clarification to improve the focus of your article.

1.     Think about including a figure that illustrates the study's methodology. It may make it easier for readers to comprehend your article's main plot.

2.     Give further information about the methodology used to measure the TEER values and whether any validation was carried out to guarantee the monolayer integrity was consistent throughout the studies.

3.     Could you elaborate on the specifics of the experimental setup used in the rat and human liver microsome metabolic stability tests? In particular, what were the concentrations and incubation durations employed, and how did these parameters correspond with expected clinical exposures?

4.     How was the dosage of tariquidar for both oral and intravenous injection determined in the tariquidar experiments? Did dose-response studies verify that the chosen dose was the most effective in blocking P-gp?

5.     How does the role of P-gp, UGT enzymes, and CYP relate to the clinical environment based on in vitro research? Are there any clinical studies or literature that either confirm or refute your findings, especially with regard to P-gp-mediated DDIs?

6.     Given the findings that P-gp significantly affects the oral bioavailability of SGC003F, are there any considerations for modifying the formulation of SGC003F to mitigate this effect?

7.     You mention potential bias in extrapolating in vitro results to in vivo situations, especially for transporter-mediated DDIs. Can you elaborate on the specific biases or limitations observed in your study, and how these might impact the interpretation of the results?

8.     The increase in tissue exposure, particularly in the brain and small intestine, raises questions about the potential for adverse effects. How do you assess the risk of toxicity in these tissues, and what are the clinical implications of these findings?

Author Response

Dear editor and reviewers:

Thank you very much for taking the time to review our manuscript “Inhibition of P-glycoprotein asymmetrically alters the in vivo exposure profile of SGC003F: a novel Guanylate Cyclase Stimulator” and provide valuable feedback. We have carefully addressed your comments and made corresponding revisions to the manuscript. Below, we provide a point-by-point response to your comments. The corresponding corrections had been highlighted in the revised manuscript.

  1. Think about including a figure that illustrates the study's methodology. It may make it easier for readers to comprehend your article's main plot.

Reply:

Thanks for your professional advice. The technical roadmap has been made and incorporated in the Figure 1B.

  1. Give further information about the methodology used to measure the TEER values and whether any validation was carried out to guarantee the monolayer integrity was consistent throughout the studies.

Reply:

Thanks for your suggestion. We have added it in the revised draft (4.3.1).

For electrical measurements, two electrodes are used, with one electrode placed in the upper compartment and the other in the lower compartment and the electrodes are separated by the cellular monolayer. The measurement procedure includes measuring the blank resistance (Rblank) of the semipermeable membrane only (without cells) and measuring the resistance across the cell layer on the semipermeable membrane (Rtotal). TEER values are achieved by calculating as TEER = (Rtotal-Rblank) × Marea in unit of Ω.cm2. The monolayer integrity was guaranteed by determining the TEER before and after the bidirectional transport experiments.

  1. Could you elaborate on the specifics of the experimental setup used in the rat and human liver microsome metabolic stability tests? In particular, what were the concentrations and incubation durations employed, and how did these parameters correspond with expected clinical exposures?

Reply:

Thank you for your valuable suggestion. We have incorporated your suggestion into the revised draft (section 4.3.2). Here are the specific changes:

The final concentration of each component in the incubation system has been stated in the article. The final concentration of SGC003F is 1 μM, which is lower than the peak concentration of SGC003F and approaches its steady-state concentration. The total incubation period was 60 min, with sampling conducted at 0, 5, 15, 30, and 60 minutes to observe the drug’s changes over time and to calculate its in vitro half-life. Positive probe substrate assays were utilized to ensure normal activity of the incubation system.

The findings of this research indicate that the compound SGC003F exhibited minimal metabolism in liver microsomes in vitro, to the extent that its hepatic clearance was incalculable. This implies a negligible contribution of hepatic metabolism to the overall clearance of SGC003F.

  1. How was the dosage of tariquidar for both oral and intravenous injection determined in the tariquidar experiments? Did dose-response studies verify that the chosen dose was the most effective in blocking P-gp?

Reply:

Thanks for your inquiry. In our experiments involving the concurrent administration of tariquidar, we primarily relied on dosages documented in the literature, which have been demonstrated to effectively inhibit P-glycoprotein (P-gp) function in rats through both intravenous and oral routes(Pharmacokinetics of the P-gp Inhibitor Tariquidar in Rats After Intravenous, Oral, and Intraperitoneal Administration. Eur J Drug Metab Pharmacokinet 2018, 43, 599-606, doi:10.1007/s13318-018-0474-x; A phase I study of the P-glycoprotein antagonist tariquidar in combination with vinorelbine. Clinical Cancer Research 2009, 15, 3574-3582.)

The pharmacokinetic (PK) data from this study reveal that the selected tariquidar dosage significantly enhanced the oral bioavailability of SGC003F, reaching an impressive 95.1%. This outcome confirmed the efficacy of the dose in blocking P-gp activity. We have expanded upon this point in the discussion section of the revised manuscript. 

  1. How does the role of P-gp, UGT enzymes, and CYP relate to the clinical environment based on in vitro research? Are there any clinical studies or literature that either confirm or refute your findings, especially with regard to P-gp-mediated DDIs?

Reply:

Thanks for your question. P-gp, UGT and CYP are important proteins that regulate drug transport and metabolism in vivo. A large amount of data and studies have demonstrated that P-gp, UGT, and CYP are key factors in causing DDI. Considering this, the FDA release a DDI research guideline (Guidance for Industry: In Vitro Drug Interaction Studies —Cytochrome P450 Enzyme-and Transporter-Mediated Drug Interactions 2020, FDA) to aid drug development. These guidelines address the impact of drug-metabolizing enzymes, predominantly the CYPs, as well as the role of transporters, with an emphasis on P-gp. The literature is replete with studies on DDIs induced by P-gp, highlighting its significance in the field of pharmacology and drug safety (Enhancement effect of P-gp inhibitors on the intestinal absorption and antiproliferative activity of bestatin. European journal of pharmaceutical sciences 2013, 50, 420-428.; P-glycoprotein inhibition increases the brain distribution and antidepressant-like activity of escitalopram in rodents. (Neuropsychopharmacology 2013, 38, 2209-2219.). In this study, in vitro metabolic stability experiments revealed that SGC003F demonstrated stability in the presence of both UGT and CYP enzymes. This finding suggests that these enzymes are not the primary mediators of SGC003F's metabolic clearance in vivo, thereby reducing the likelihood of DDIs associated with metabolic pathways. However, trans-membrane transport studies indicated that SGC003F is a substrate for P-gp, elevating the potential for P-gp-mediated DDIs. Following co-administration with tariquidar, a P-gp inhibitor, the bioavailability of SGC003F increased to 95%, indicating that affecting the function of P-gp can cause DDI. These insights have been elaborated upon in the discussion section of the revised draft to address the implications of P-gp in the context of DDIs for SGC003F.

  1. Given the findings that P-gp significantly affects the oral bioavailability of SGC003F, are there any considerations for modifying the formulation of SGC003F to mitigate this effect?   

Reply:

Thanks for your suggestion. From the perspective of new drug development, a strategic focus will be placed on mitigating the impact of P-gp through the optimization of the prescription formulation. This approach is considered, where feasible, during the early stages of formulation development for SGC003F.

  1. You mention potential bias in extrapolating in vitro results to in vivo situations, especially for transporter-mediated DDIs. Can you elaborate on the specific biases or limitations observed in your study, and how these might impact the interpretation of the results?

Reply:

Thanks for your question. Transporter-based in vitro experiments to predict DDI in vivo started later than DDI prediction based on metabolic enzymes CYP and UGT, and the related success experience is still limited. In the case of the present study, in the in vitro transmembrane transport experiments, SGC003F was demonstrated to be a substrate for P-gp. However, the kinetic parameters, namely Vmax and Km, which describe the binding of SGC003F to P-gp, have not been established. The extrapolation of these in vitro findings to predict in vivo outcomes is currently constrained by a lack of comprehensive data, which introduces potential for bias. Further research is necessary to refine our understanding and predictions of transporter-mediated DDIs.

  1. The increase in tissue exposure, particularly in the brain and small intestine, raises questions about the potential for adverse effects. How do you assess the risk of toxicity in these tissues, and what are the clinical implications of these findings?

Reply:

Thanks for your inquiry. The findings of our study indicate that the co-administration of P-gp inhibitors leads to a significant increase in the blood concentrations of SGC003F, which is in line with the enhanced drug exposure observed in most tissues. Notably, the exposures in the intestine and brain are substantially higher compared to other tissues. These results underscore the importance of our findings for preclinical safety and toxicological assessments, particularly in highlighting the need to consider the potential toxicity of the drug to the central nervous system and gastrointestinal tract. To address this, we have included additional relevant data and discussion in our revision to provide a more comprehensive understanding of the implications for drug safety and toxicity.

Finally, we hope that these revisions address your concerns and improve the quality of the manuscript. If you have any further comments or questions, please do not hesitate to contact us. We are grateful for your constructive feedback and appreciate your assistance in enhancing the clarity and accuracy of our work.

Best regards,

Sincerely Yours,

Xiaomei Zhuang

State Key Laboratory of Toxicology and Medical Countermeasures, Beijing Institute of Pharmacology and Toxicology, Beijing, China

[email protected]

Reviewer 2 Report

Comments and Suggestions for Authors

The article “Inhibition of P-glycoprotein asymmetrically alters the in vivo exposure profile of SGC003F: a novel Guanylate Cyclase Stimulator” is interesting, overall, it is well written, but I have following comments and suggestions,

1.      The abstract must contain information on SGC003F, where it is indicated?

2.      The introduction is well-written, and the aim is clear.

3.      The  International Transporter Consortium (ITC) has stated in its guidelines and established decision trees that, when the net flux ratio of a drug in a bidirectional transport assay is less than 2, then it is a poor or non-P-gp substrate, how do the authors comment on it?

4.      I can't see details about the calculation of ke in the program Phoenix used for non-compartmental analysis, as there are some PK profiles in Figure 4 where the method for calculating Ke can affect the calculated PK=parameters.

Author Response

(The authors gave the same response as above.)

Reviewer 3 Report

Comments and Suggestions for Authors

The authors performed a very complete in vitro and in vivo pharmacokinetic investigation of SGC003F together tariquidar, a P-gp inhibitor. The study conducted numerous experiments, including metabolic stability assays using rat and human liver microsomes, bidirectional transmembrane transport experiments utilizing Caco-2 cells, LLC-PK-MOCK, and LLC-PK1-MDR1 cells, as well as in vivo studies at the animal level. Additionally, tissue distribution and excretion experiments were performed to assess the pharmacokinetics of SGC003F. I have only some considerations before the acceptance of the manuscript in Pharmaceuticals:

(1) The quality of Figure 1 must be improved. Please use professional software, such as ChemDraw, for drawing the chemical structures. In others figures, please check carefully the font size in the graph axis.

(2) The abbreviation "DDI" must be defined the first time it appears in the text.

(3) The abstract needs to include the most important numerical values of the pharmacokinetic data for clarity and specificity.

(3) The introduction does not describe the structure of riociguat. Please include a detailed description of riociguat's chemical structure.

(4) Riociguat is primarily used for treating pulmonary hypertension. Its potential use in heart failure should be better contextualized within the introduction.

(5) Stating that compounds "were synthesized in-house" is not sufficient. Please cite relevant references with spectroscopic data or provide these information (at least 1H and 13C NMR) to validate the identity of the compound SGC003F. This is the most important issue to be considered by the authors.

(7) Ensure all Latin names in the text are italicized.

(8) The use of ANOVA for mean comparison has been mentioned, but information about the adherence of the data to the normal distribution has not been described. Please provide details on how normality was tested and confirmed.

Author Response

Thank you very much for taking the time to review our manuscript “Inhibition of P-glycoprotein asymmetrically alters the in vivo exposure profile of SGC003F: a novel Guanylate Cyclase Stimulator” and provide valuable feedback. We have carefully addressed your comments and made corresponding revisions to the manuscript. Below, we provide a point-by-point response to your comments.The corresponding corrections had been highlighted in the revised manuscript.

  • The quality of Figure 1 must be improved. Please use professional software, such as ChemDraw, for drawing the chemical structures. In others figures, please check carefully the font size in the graph axis.

Reply:

Thanks for your suggestion. The revisions have been incorporated into Figure 1 of the updated manuscript, and similar updates have been made to the other figures as well.

  • The abbreviation "DDI" must be defined the first time it appears in the text.

Reply:

Thank you for reviewing. Addition has been made in the revised draft (conclusions section in the abstract).

  • The abstract needs to include the most important numerical values of the pharmacokinetic data for clarity and specificity

Reply:

Thank you for your insightful suggestion. We have made necessary revision in the results section of the abstract. The inclusion of tariquidar significantly altered the pharmacokinetics of SGC003F. In LLC-PK1-MDR1 cells, tariquidar reduced the efflux ratio of SGC003F from 6.56 to 1.28. In rats, it enhanced the plasma AUC by 3.05 or 1.61 times, increased the Cmax by 2.13 or 1.07 times, and notably improved bioavailability from 46.4% to 95%. Additionally, co-administration with tariquidar led to a decrease in fecal excretion and an increase in tissue exposure, with only a moderate effect on the partition ratios in the small intestine and brain.

  • The introduction does not describe the structure of riociguat. Please include a detailed description of riociguat's chemical structure.

Reply:

Thanks for your question. Riociguat serves as an internal standard and its chemical structure is depicted in Figure 1 of the revised manuscript.

  • Riociguat is primarily used for treating pulmonary hypertension. Its potential use in heart failure should be better contextualized within the introduction.

Reply:

Thanks for your valuable advice. We have added this background in the introduction section of the updated manuscript.

Recent studies have shown that nitric oxide (NO) -soluble guanylate cyclase (sGC) -cyclic guanosine monophosphate (cGMP) pathway plays a crucial role in regulating cardiovascular function and serve as a novel target for the treatment of HF. Riociguat, the inaugural sGC stimulator approved for clinical use, has seen restricted application in HF treatment due to its brief half-life, a consequence of metabolism by cytochrome P450 (CYP) enzymes. Primarily, it is now prescribed for pulmonary arterial hypertension and chronic thromboembolic pulmonary hypertension. The development of Vericiguat, a derivative of riociguat with an extended half-life achieved through structural modification, marks a significant advancement. Vericiguat is the first-in-class drug that targets the NO-sGC-cGMP signaling pathway for the treatment of heart failure with reduced ejection fraction (HFrEF). Clinical studies have demonstrated that the adjunctive use of Vericiguat with standard therapy can lead to a substantial reduction in the risks of cardiovascular mortality or heart failure-related hospitalization, offering a promising new strategy in the management of heart failure.  

  • Stating that compounds "were synthesized in-house" is not sufficient. Please cite relevant references with spectroscopic data or provide these information (at least 1H and 13C NMR) to validate the identity of the compound SGC003F. This is the most important issue to be considered by the authors.

Reply:

Thanks for your question. We have added the proof of authorized patent for SGC003F in Revision 4.1, which contains the NMR spectrum results of SGC003F.(Zheng, Z.; Zhang, Y.; Li, S.; Zhuang, X.; Li, S.; Li, P.; Cai, X.; Xiao, J.; Li, X. Substituted Thiophene-5-fluoro-1H-Pyrazolo[3,4-d]Pyrimidine Compounds and Their Applications. CN117924280B, 2024-07-12.)

  • Ensure all Latin names in the text are italicized.

Reply:

Thanks for your suggestion. All the Latin names involved in the manuscript had been presented by italicized pattern.

  • The use of ANOVA for mean comparison has been mentioned, but information about the adherence of the data to the normal distribution has not been described. Please provide details on how normality was tested and confirmed.

Reply:

Thank you for your insightful suggestion. We have included the normal distribution testing method in section 4.6 of the revised draft.

Statistical analyses were performed using IBM SPSS Statistics 26. The experimental data were expressed as mean ± standard deviation (SD). Shapiro-Wilk test was firstly utilized to test the normal distribution of the data. When exploring differences between multiple groups, a one-way ANOVA was used to determine if there was a significant difference between group means. For the comparison of the two groups of data, the independent sample t-test was used for statistical analysis. P value less than 0.05 was set as the threshold of significance. When p < 0.05, the observed difference was judged to be statistically significant.

We hope that these revisions address your concerns and improve the quality of the manuscript. If you have any further comments or questions, please do not hesitate to contact us. We are grateful for your constructive feedback and appreciate your assistance in enhancing the clarity and accuracy of our work.

Best regards,

Sincerely Yours,

Xiaomei Zhuang

State Key Laboratory of Toxicology and Medical Countermeasures, Beijing Institute of Pharmacology and Toxicology, Beijing, China

[email protected]

Reviewer 4 Report

Comments and Suggestions for Authors

The study evaluates potential implication of P-gp in pharmacokinetics of SGC000F, using tariquidar as a P-gp inhibitor.

Comments:

Introduction

The paragraph depicting treatment approach to heart failure is not needed.

Page 2 line 3 down: after (cGMP) „pathway” should be added

Page 2 vericiguat was proven of therapeutic value in some specific clinical conditions, not general in heart failure, this should be specified

Page 2 line 15 top. summary of product characteristics state that vericiguat is a substrate of P-gp (and BCRP), but with low interaction potential, and contrary to the information provided in the introduction published studies (e.g.  Clin Transl Sci. 2023 Dec;16(12):2458-2466. doi: 10.1111/cts.1367 – review; Clin Pharmacokinet. 2020 Nov;59(11):1407-1418. doi: 10.1007/s40262-020-00895-x – original), this facts should be presented in the introduction.

Some comments about tariquidar should be presented in the introduction, and there some data that the agent is also substrate/inhibitor of other transporters like BCRP and MRP1.

Methods

It would be advisable weather SGC000F is a substrate of any active transporter, i.e. determination of active transport involvement in transmembrane shift of the agent in cells cultured at 4°C.

Results

To be considered to place the tables (table 1 and 2) in the supplementary file, as duplication with figure (figures 2 and 3) data appear.

Discussion

BCRP is also expressed in the intestine, and facing information that tariquidar might be its inhibitor the contribution of this transporter could be debated (the same for BBB barrier and kidney, but rather not liver, as BCPR expression in this tissue is not high -  Clin Pharmacol Ther. 2019 May;105(5):1204-1212. doi: 10.1002/cpt.1301).

Some comments to the concentrations used, as these were of microM, which are higher that those observed in clinical life.

Author Response

Thank you very much for taking the time to review our manuscript “Inhibition of P-glycoprotein asymmetrically alters the in vivo exposure profile of SGC003F: a novel Guanylate Cyclase Stimulator” and provide valuable feedback. We have carefully addressed your comments and made corresponding revisions to the manuscript. Below, we provide a point-by-point response to your comments.

Introduction 

  1. The paragraph depicting treatment approach to heart failure is not needed.

Reply:

Thanks for your advice. The section describing the approach to heart failure was deleted upon the introduction to the revised manuscript.

Heart failure (HF) is a significant global public health issue characterized by high morbidity and mortality. HF refers to the inability of the heart to pump enough blood to maintain the body's metabolism needs, leading to symptoms such as dyspnea, limited activity, and fluid retention. In severe cases, symptoms such as shortness of breath, chest tightness, double leg edema, dizziness, and may even be fatal.

  1. Page 2 line 3 down: after (cGMP) „pathway” should be added

Reply:

Thanks for your suggestion, we have added “pathway” after cGMP of the revised version. 

Recent studies have shown that nitric oxide (NO) -soluble guanylate cyclase (sGC) -cyclic guanosine monophosphate (cGMP) pathway plays a crucial role in reg-ulating cardiovascular function and serve as a novel target for the treatment of HF.

3.Page 2 vericiguat was proven of therapeutic value in some specific clinical conditions, not general in heart failure, this should be specified.

Reply:

Thank you for your valuable advice. It has been supplemented in the introduction section.

Recent studies have shown that nitric oxide (NO) -soluble guanylate cyclase (sGC) -cyclic guanosine monophosphate (cGMP) pathway plays a crucial role in regulating cardiovascular function and serve as a novel target for the treatment of HF. The first sGC stimulator approved for clinical use was riociguat; however, its short half-life, attributed to metabolic effects by CYP enzymes, limited its application in the field of heart failure. Currently, riociguat is primarily utilized for the treatment of pulmonary hypertension and chronic thromboembolic pulmonary hypertension. Riociguat underwent structural modification to yield Vericiguat, which extends its half-life and represents the first drug used to treat HFrEF by stimulating the NO-sGC-cGMP signaling pathway. Currently, clinical use of Vericiguat is indicated for the treatment of adults with symptomatic chronic heart failure with an EF of less than 45% following hospitalisation for heart failure or the need for outpatient intravenous diuretics. Studies have shown that the combined use of Vericiguat with standard therapy can significantly reduce the risk of cardiovascular death or hospitalization for heart failure in patients, providing a new approach to the treatment of heart failure.  
4.Page 2 line 15 top. summary of product characteristics state that vericiguat is a substrate of P-gp (and BCRP), but with low interaction potential, and contrary to the information provided in the introduction published studies (e.g.  Clin Transl Sci. 2023 Dec;16(12):2458-2466. doi: 10.1111/cts.1367 – review; Clin Pharmacokinet. 2020 Nov;59(11):1407-1418. doi: 10.1007/s40262-020-00895-x – original), this facts should be presented in the introduction.

Reply:

Thanks for your suggestion. We have incorporated the necessary information from the literature. Vericiguat is a substrate of P-gp and BCRP. However, no clinically significant differences were observed with co-administration of digoxin (P-gp substrate) (FDA Approved Drug Products: Verquvo (vericiguat) oral tablets).

5.Some comments about tariquidar should be presented in the introduction, and there some data that the agent is also substrate/inhibitor of other transporters like BCRP and MRP1.

Reply:

Thanks for your value advice. In the introduction section, we have supplemented a background on tariquidar, a potent, specific, noncompetitive P-glycoprotein inhibitor. Previous studies have reported that the pharmacokinetics of many drugs may be affected when co-administered with tariquidar. For example, the effect of tariquidar co-administration with loperamide on the pharmacokinetics and brain distribution of loperamide in rats. One hour after administration, 1.0 mg/kg of tariquidar increased loperamide levels in the brain by 2.3-fold. Regarding the influence of tariquidar on other transporters such as BCRP and MRP1, we have included a detailed discussion in the corresponding section of our paper.

Methods 

6.It would be advisable weather SGC000F is a substrate of any active transporter, i.e. determination of active transport involvement in transmembrane shift of the agent in cells cultured at 4°C.

Reply:

Thanks for your suggestion. In this study, we performed bidirectional SGC00F transport experiments using two distinct cell lines: LLC-PK1-MDR1, which overexpresses P-glycoprotein (P-gp), and its control counterpart, LLC-PK1-MOCK. The findings from these experiments not only confirmed that SGC003F is a specific substrate for P-gp, but also demonstrated its moderate membrane permeability. This study serves as an initial exploration into the role of P-gp in the transport of SGC003F. Moving forward, we are committed to further investigating the influence of additional transporters on SGC003F in greater detail.

Results 

  1. To be considered to place the tables (table 1 and 2) in the supplementary file, as duplication with figure (figures 2 and 3) data appear.

Reply:

Thanks for your insightful suggestion. We have decided to remove Table 1 to the supplementary file. However, we believe it would be more beneficial to retain Table 2 within the main body of the text. The rationale for this decision is that Figure 4 primarily illustrates the plasma concentration-time curves for various delivery modes, both with and without the presence of an inhibitor. Critical pharmacokinetic parameters such as bioavailability, clearance, and half-life are not depicted in Figure 4. By including Table 2 in the main text, readers can conveniently compare the ratios of key pharmacokinetic parameters, including bioavailability, plasma exposure, and clearance, between conditions with and without the inhibitor.

Discussion 

8.BCRP is also expressed in the intestine, and facing information that tariquidar might be its inhibitor the contribution of this transporter could be debated (the same for BBB barrier and kidney, but rather not liver, as BCPR expression in this tissue is not high -  Clin Pharmacol Ther. 2019 May;105(5):1204-1212. doi: 10.1002/cpt.1301). Some comments to the concentrations used, as these were of microM, which are higher that those observed in clinical life.

Reply:

Thank you for your suggestion. Although tariquidar is widely used as a inhibitor of P-gp both in vitro and in vivo studies in animals, recent studies have found that it still suffers from poor specificity. This lack of specificity is one of the challenges that hinder the study of transporters. The concentrations of tariquidar used in vitro and in vivo experiments in this study were designed according to the literature reports. However, we acknowledge that the absence of tariquidar concentration verification is a limitation of this study. Additionally, we cannot rule out the potential involvement of BCRP in the transport of SGC003F. We have provided further clarification and discussion on these points in the 'Discussion' section.

We hope that these revisions address your concerns and improve the quality of the manuscript. We believe this revised text provides a more structured and focused presentation of the story. If you have any further comments or questions, please do not hesitate to contact us. Finally, we are grateful for your constructive feedback and appreciate your assistance in enhancing the clarity and accuracy of our work.

Best regards,

Sincerely Yours,

Xiaomei Zhuang

State Key Laboratory of Toxicology and Medical Countermeasures, Beijing Institute of Pharmacology and Toxicology, Beijing, China

[email protected]

Round 2

Reviewer 1 Report

Comments and Suggestions for Authors

Thank you for your insightful clarifications. Your explanations have significantly improved the clarity of the article, making it easier to understand from my perspective.

Reviewer 4 Report

Comments and Suggestions for Authors

Accept in present form